# Morphometric Parameters of Krumbein Grain Shape Charts— A Critical Approach in Light of the Automatic Grain Shape Image Analysis

**Jacek Bogusław Szmańda *** 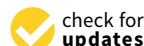 **and Karol Witkowski**

Department of Geoinformation and Environmental Research, Institute of Geography, Pedagogical University of Krakow, Podchorążych 2, 30-084 Kraków, Poland; karol.witkowski@up.krakow.pl
* Correspondence: jacek.szmanda@up.krakow.pl

**Abstract:** Grain-shape analyses are essential in geological research because they provide the basis for genetic interpretations, including sedimentation conditions. The methods of visual evaluation used so far have been subjective, time-consuming and labour intensive. Automatic particle image analysis, including the methods used by the Morphology G3SE device, open up the possibility of mass and objective roundness analysis of mineral and organic particles. The article presents the results of measurements for the grain scale proposed by Krumbein in 1941, as this scale has been used in numerous sedimentological studies. The standard shapes were analysed using four parameters: High Sensitivity (HS) Circularity, Convexity, Solidity and Aspect Ratio. In the discussion, both the results and the grain-shape standards were critically assessed. The most important trends in the distribution of morphometric parameters of the scale are shown. On this basis, it was found that it is impossible to determine the parameter boundary values that would distinguish each class of grain roundness proposed by Krumbein. The conclusions propose criteria for the automatic differentiation of angular, subrounded and rounded grains, which could be a basis for describing the shape of mineral particles.

**Keywords:** textural analysis; automatic grain shape analysis; pattern of particle shape; roundness

## 1. Introduction

Geological textural analyses consist of determining grain size and shape. Grain shape, fundamental in studying sedimentation environments [1], is determined using many methods and measurement techniques. These include methods based on both grain-shape measurements [2,3] and visual evaluation, as proposed by Krumbein [4], Powers [5] and Cailleux [6,7]. The visual assessment of the grain shape is laborious and subjective, but it is widespread in the study of the characteristics of rocks and their genesis [8–17]. Genetic interpretation of sediments is often based on the results of sand (quartz) grain-coating analyses. Textural maturity of sediments depends on the shape of grains [18]. This shape, however, depends on the characteristics of the source material and the transport distance [7,19,20]. Regardless of the characteristics of the sedimentation medium (air or water), roundness increases with distance of grain transport [21]. Therefore, in interpreting the origin of sediments, the analysis of the shape of sand grains is essential. However, it should be emphasised that grain shapes in various sedimentary environments are similar, especially fluvial, aeolian and beach sediments, as well as glacial and weathered [14,19].

Today, several available devices automatically and objectively analyse the shape of grains [14–16,22–25]. One of them is the Morphologi G3SE. The algorithm of this computer program automatically analyses the two-dimensional digital image. As a result of automatic measurements, grain-size distribution (quantitative and volumetric) is obtained, and 17 parameters of the shape of each grain are calculated [26,27].

As of yet, only a few studies of the shape of mineral particles from various sedimentary environments have been carried out with the Morphologi G3SE apparatus [28–34]. According to Resentini et al. [31] and Varga et al. [33], despite the technological advances made in computer-image analysis, no entirely satisfactory automatic method has yet been invented to quantify the shape of detritic particles, which would enable the unambiguous identification of their genesis.

Grain-shape studies conducted with time-consuming subjective methods are often used in the interpretation of sediment genesis. Along with developing automatic grain-image analyses, the opportunity to compare the results of morphometric measurements using these methods with the grain-shape scale proposed by Krumbein [4] has arisen.

An automated quantitative analysis would eliminate the subjective judgment that is unavoidable with non-automated visual analysis. So far, no values for specific parameters calculated by the Morphologi G3SE software, which would allow for distinguishing angular, subrounded and rounded grains, have been determined. Because Krumbein's chart [4] is a model that has repeatedly constituted the basis for distinguishing types of grain roundness, the question arises if it is possible to automate the analysis of Krumbein's patterns using measurement data from the Morphologi G3SE device. Answering this was the main task of the research. Its most important goal was to define criteria for separating angular, subrounded and rounded grains with morphometric parameter values. The article also forms a part of the discussion that has been going on for many years on the significance of Krumbein grain-shape classes when analysing the coating of mineral clusters (mainly quartz sand grains) [7,8,14,19,28,31,34–36].

## 2. Materials and Methods

When analysing the shape of mineral particles using Krumbein's chart [4], three types of grain rounding can be distinguished. Each of these classes is divided into three and found on a nine-point scale from 0.1 to 0.9 (Figure 1): (1) angular grains: 0.1–0.3; (2) subrounded grains: 0.4–0.6; and (3) rounded grains: 0.7–0.9. The tenth group is broken grains p1–6. The three groups on Krumbein's chart formed the scale used to distinguish several types corresponding to different sedimentation environments [7,19,35]: (1) 0.1–0.2 roundness of grains according to Krumbein—NU glacial and weathering—fresh and angular [7]; (2) 0.3–0.6: EM glaciofluvial, aeolian and beach—moderately rounded [7]; (3) 0.7–0.9: EL fluvial and beach—well rounded smooth and shiny surface, and RM aeolian—well rounded with a matte surface [7]. Cracked grains were not included in the research.

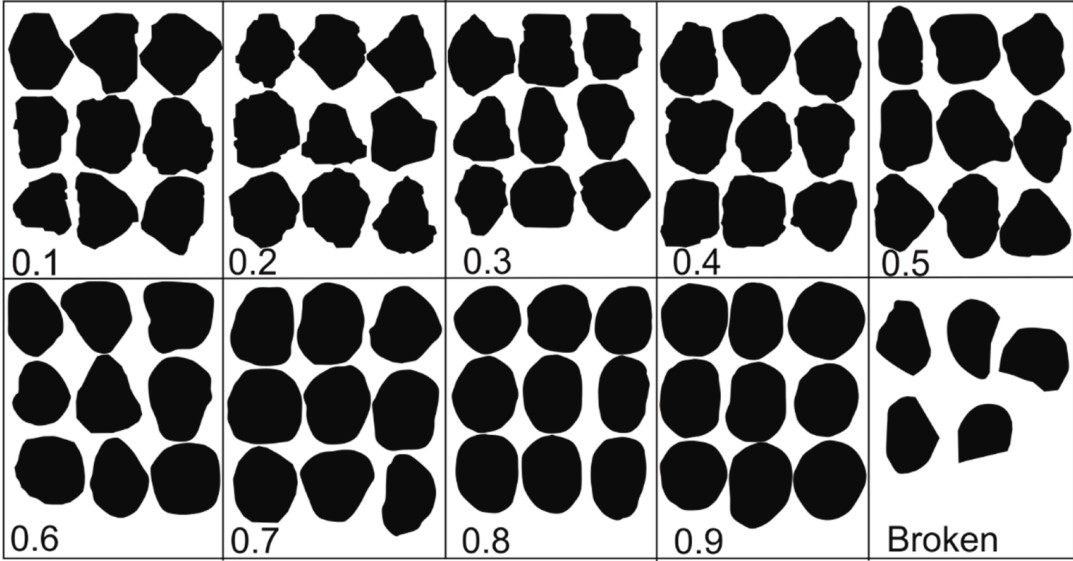

**Figure 1.** Patterns of grain roundness classes by Krumbein's chart [4].

The particle shape analysis was performed by measuring an image of standard grains using the measure-image file function. An image in jpg format with a resolution of 600 dpi was imported to the program. When measuring particle size from the image, the program is based on pixel edge length. Since the particle size for this study was not significant, pixel size was set at 1 micron. Then, the threshold value was determined, allowing a preprocessing stage to remove "virtual contaminants" from the image of the tested particle from the measuring plate or on the measured graphic file. When measuring an image of a particle imported in a graphic file, the threshold value is precisely set at the interface between the grain drawing (black and shades of grey) and the white background. In this study, the value was 130. After completing the measurement settings, the program automatically performs the analysis.

Out of 17 morphometric parameters generated by the Morphologi G3SE program, four were used in the study. The choice of these parameters resulted from their importance in comprehensive analyses of shapes previously performed with this device [28–34,36]. Similar parameters were also used in other grain-coating studies [14,28,37,38].

The following parameters were used: (1) HS Circularity, (2) Convexity, (3) Solidity, (4) Aspect Ratio. Their characteristics are described below, based on the Malvern manual for Morphologi G3SE (Malvern Panalytical Ltd., Cambridge, UK) [27].

HS Circularity (High Sensitivity Circularity) is the ratio of the circumference of a circle, equal to the object's projected area, to the perimeter of the grain. A perfect grain circle has an HS Circularity of 1.0, while a narrow rod particle has an HS Circularity close to 0. It is calculated (Formula (1)) [27]:

$$\text{HS Circularity} = (4 \times \pi \times \text{Area}) / \text{Perimeter}^2 \tag{1}$$

Convexity is the perimeter of the convex hull perimeter of the particle (A + B) divided by its perimeter (A) (Figure 2). Its values range from 0 for least convex, very spiky and irregular particles, to 1 for most convex, smooth shape particles. Convexity is calculated as (Formula (2)) [27]:

$$\text{Convexity} = (\text{Perimeter of A} + \text{B}) / (\text{Perimeter of A}) \tag{2}$$

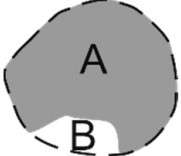

**Figure 2.** Scheme of a particle: the gray area (A) and hull-dotted line (A + B) [27].

Solidity is the grain area (A) divided by the area enclosed by the convex hull (A + B) The grains with the Solidity value of 1 are characterised by smoothness, while the values of 0 are particles with irregular and rough edges. Solidity is calculated as (Formula (3)) [27]:

$$\text{Solidity} = (\text{Area of A}) / (\text{Area of A} + \text{B}) \tag{3}$$

Aspect Ratio is the ratio of the width to length of the particle. Aspect Ratio values range from 0 to 1. For example, the Aspect Ratio of a circle has a value of 1. Aspect Ratio is calculated as (Formula (4)) [27]:

$$\text{Aspect Ratio} = (\text{Width}) / (\text{Length}) \tag{4}$$

The average value (AVG) and standard deviation (SD) were calculated for each Krumbein grain class by analysing the parameters. In addition, an anomaly analysis was performed, determining typical values for each class, ranging from AVG − SD to AVG + SD, with anomalous values being outside this range. If the value was lower than AVG − SD, it was marked as a negative anomalous value (A−). However, if this value was higher than

AVG + SD, it was positively anomalous (A+). The relationships between the indicators were determined by calculating the Pearson Correlation Coefficient (PCC).

## 3. Results

The numerical values of the morphometric parameters of the classes analysed are presented in Table 1. When analysing the values of the parameters for each class proposed by Krumbein (Table 1), general growth in the HS Circularity, Convexity and Solidity parameters can be observed with an increase in the rounding class (Figure 3). This relationship manifests itself in a strong positive correlation between the values of the indicators: (1) HS Circularity and Convexity—0.9 PCC; (2) HS Circularity and Solidity—0.97 PCC; (3) Convexity and Solidity—0.96 PCC. It is also clearly visible in the diagrams of the relationships of the examined parameters (Figure 4). However, the relationship between the values of these indicators and Aspect Ratio values is very weak: (1) HS Circularity and Aspect Ratio—0.32 PCC; (2) Convexity and Aspect Ratio—0.17 PCC; (3) Solidity and Aspect Ratio—0.22 PCC.

**Table 1.** Values of morphometric parameters of Krumbein's grain shape charts.

| Class. No. | Shape Pattern | HS Circularity | Convexity | Solidity | Aspect Ratio |
|---|---|---|---|---|---|
| 1.1 | | 0.895 A+ | 0.995 A+ | 0.979 | 0.818 |
| 1.2 | | 0.801 | 0.98 | 0.953 A− | 0.896 |
| 1.3 | | 0.87 | 0.99 | 0.97 A+ | 0.965 |
| 1.4 | | 0.818 | 0.974 | 0.963 | 0.749 |
| 1.5 | | 0.862 | 0.974 | 0.972 | 0.818 |
| 1.6 | | 0.843 | 0.964 − | 0.962 | 0.887 |
| 1.7 | | 0.771 A− | 0.957 A− | 0.948 A− | 0.985 |
| 1.8 | | 0.815 | 0.984 | 0.966 | 0.822 |
| 1.9 | | 0.872 | 0.987 | 0.982 | 0.783 |
| AVG | | 0.855 | 0.978 | 0.966 | 0.858 |
| SD | | 0.040 | 0.012 | 0.011 | 0.081 |
| 2.1 | | 0.775 A− | 0.943 A− | 0.949 | 0.795 |

**Table 1.** *Cont.*

| Class. No. | Shape Pattern | HS Circularity | Convexity | Solidity | Aspect Ratio |
|:---:|:---:|:---:|:---:|:---:|:---:|
| 2.2 | | 0.838 | 0.969 | 0.955 | 0.886 |
| 2.3 | | 0.809 | 0.989 A+ | 0.965 | 0.889 |
| 2.4 | | 0.838 | 0.959 | 0.974 | 0.896 |
| 2.5 | | 0.787 | 0.971 | 0.936 A− | 0.883 |
| 2.6 | | 0.863 | 0.985 | 0.962 | 0.982 |
| 2.7 | | 0.901 A+ | 0.987 | 0.975 A+ | 0.925 |
| 2.8 | | 0.878 A+ | 0.978 | 0.973 | 0.856 |
| 2.9 | | 0.755 A− | 0.953 A− | 0.938 A− | 0.836 |
| AVG | | 0.827 | 0.970 | 0.959 | 0.883 |
| SD | | 0.049 | 0.016 | 0.015 | 0.053 |
| 3.1 | | 0.847 | 0.982 | 0.959 A− | 0.832 |
| 3.2 | | 0.815 A− | 0.975 A− | 0.966 | 0.791 |
| 3.3 | | 0.879 | 0.981 | 0.97 | 0.854 |
| 3.4 | | 0.83 | 0.992 | 0.964 | 0.907 |
| 3.5 | | 0.853 | 0.994 | 0.983 | 0.65 |
| 3.6 | | 0.865 | 0.994 | 0.979 | 0.704 |
| 3.7 | | 0.838 | 0.977 A− | 0.963 | 0.746 |
| 3.8 | | 0.948 A+ | 0.999 A+ | 0.990 A+ | 0.993 |

**Table 1.** *Cont.*

| Class. No. | Shape Pattern | HS Circularity | Convexity | Solidity | Aspect Ratio |
|:---:|:---:|:---:|:---:|:---:|:---:|
| 3.9 | | 0.906 | 0.99 | 0.990 A+ | 0.976 |
| AVG<br>SD | | 0.865<br>0.041 | 0.987<br>0.009 | 0.974<br>0.012 | 0.828<br>0.650 |
| 4.1 | | 0.881 | 0.987 | 0.985 | 0.809 |
| 4.2 | | 0.905 A+ | 0.995 A+ | 0.985 | 0.844 |
| 4.3 | | 0.887 | 0.992 | 0.978 | 0.799 |
| 4.4 | | 0.846 A− | 0.972 A− | 0.965 A− | 0.905 |
| 4.5 | | 0.906 A+ | 0.986 | 0.987 | 0.807 |
| 4.6 | | 0.853 A+ | 0.983 | 0.967 A− | 0.815 |
| 4.7 | | 0.885 | 0.991 | 0.986 | 0.807 |
| 4.8 | | 0.905 A+ | 0.984 | 0.982 | 0.906 |
| 4.9 | | 0.871 | 0.987 | 0.978 | 0.855 |
| AVG<br>SD | | 0.882<br>0.022 | 0.986<br>0.007 | 0.979<br>0.008 | 0.839<br>0.042 |
| 5.1 | | 0.829 A− | 0.986 A− | 0.993 A+ | 0.57 |
| 5.2 | | 0.932 A+ | 0.998 A+ | 0.984 | 0.941 |
| 5.3 | | 0.881 | 0.993 | 0.977 | 0.746 |
| 5.4 | | 0.892 | 0.997 | 0.99 | 0.643 |
| 5.5 | | 0.885 | 0.99 | 0.969 A− | 0.787 |

**Table 1.** *Cont.*

| Class. No. | Shape Pattern | HS Circularity | Convexity | Solidity | Aspect Ratio |
|---|---|---|---|---|---|
| 5.6 | | 0.843 A− | 0.99 | 0.972 A− | 0.69 |
| 5.7 | | 0.864 | 0.993 | 0.979 | 0.769 |
| 5.8 | | 0.88 | 0.989 | 0.976 | 0.738 |
| 5.9 | | 0.905 | 0.998 | 0.991 A+ | 0.997 |
| AVG | | 0.879 | 0.993 | 0.981 | 0.765 |
| SD | | 0.031 | 0.004 | 0.009 | 0.135 |
| 6.1 | | 0.936 | 0.998 | 0.996 | 0.755 |
| 6.2 | | 0.918 | 1 | 0.995 | 0.804 |
| 6.3 | | 0.91 | 0.995 | 0.983 A− | 0.981 |
| 6.4 | | 0.939 | 0.996 | 0.988 | 0.908 |
| 6.5 | | 0.886 A− | 0.995 | 0.987 | 0.84 |
| 6.6 | | 0.918 | 0.997 | 0.989 | 0.752 |
| 6.7 | | 0.954 | 0.996 | 0.994 | 0.907 |
| 6.8 | | 0.923 | 1 | 0.997 | 0.744 |
| 6.9 | | 0.977 A+ | 1 | 1 A+ | 0.969 |
| AVG | | 0.929 | 0.997 | 0.992 | 0.851 |
| SD | | 0.026 | 0.002 | 0.006 | 0.094 |
| 7.1 | | 0.935 | 1 | 0.997 | 0.781 |
| 7.2 | | 0.956 | 1 | 0.993 | 0.782 |

**Table 1.** *Cont.*

| Class. No. | Shape Pattern | HS Circularity | Convexity | Solidity | Aspect Ratio |
|:---:|:---:|:---:|:---:|:---:|:---:|
| 7.3 |  | 0.934 | 0.994 A− | 0.987 A− | 0.88 |
| 7.4 |  | 0.978 A+ | 0.999 | 0.999 | 0.989 |
| 7.5 |  | 0.96 | 0.999 | 0.996 | 0.827 |
| 7.6 |  | 0.943 | 0.999 | 0.995 | 0.794 |
| 7.7 |  | 0.971 | 1 | 1 | 0.82 |
| 7.8 |  | 0.948 | 1 | 0.998 | 0.89 |
| 7.9 |  | 0.877 A− | 0.997 | 0.984 A− | 0.628 |
| AVG | | 0.945 | 0.999 | 0.994 | 0.821 |
| SD | | 0.030 | 0.002 | 0.005 | 0.098 |
| 8.1 |  | 0.984 | 1 | 1 | 0.936 |
| 8.2 |  | 0.973 | 0.998 A− | 0.997 A− | 0.962 |
| 8.3 |  | 0.963 | 1 | 1 | 0.809 |
| 8.4 |  | 0.975 | 1 | 0.999 | 0.852 |
| 8.5 |  | 0.968 | 1 | 1 | 0.762 |
| 8.6 |  | 0.918 A− | 0.999 | 0.997 A− | 0.657 |
| 8.7 |  | 0.968 | 0.999 | 1 | 0.799 |
| 8.8 |  | 0.971 | 1 | 1 | 0.765 |
| 8.9 |  | 0.923 A− | 1 | 0.998 | 0.671 |

**Table 1.** *Cont.*

| Class. No. | Shape Pattern | HS Circularity | Convexity | Solidity | Aspect Ratio |
|:---:|:---:|:---:|:---:|:---:|:---:|
| AVG | | 0.960 | 1.000 | 0.999 | 0.801 |
| SD | | 0.023 | 0.001 | 0.001 | 0.104 |
| 9.1 | | 0.984 | 1 | 1 | 0.934 |
| 9.2 | | 0.942 A− | 1 | 1 | 0.709 |
| 9.3 | | 0.983 | 0.999 | 0.999 | 0.962 |
| 9.4 | | 0.955 | 1 | 0.998 | 0.758 |
| 9.5 | | 0.943 A− | 1 | 0.992 A− | 0.739 |
| 9.6 | | 0.981 | 1 | 1 | 0.847 |
| 9.7 | | 0.991 | 1 | 1 | 0.972 |
| 9.8 | | 0.948 | 0.999 | 0.999 | 0.734 |
| 9.9 | | 0.99 | 1 | 1 | 0.899 |
| AVG | | 0.969 | 1.000 | 0.999 | 0.839 |
| SD | | 0.021 | 0.000 | 0.003 | 0.106 |

AVG: average, SD: standard deviation, A−: anomalous value negatively, A+: anomalous value positive.

When analysing the variability of parameter values in Figures 3 and 4 and Table 1, it can be concluded: (1) The most anomalous values of HS Circularity were found in class 2 (four values including two negatives and two positives), and class 4 (five values including two negatives and three positives). (2) The most significant differences in the values occurred, however, between the standard grains belonging to classes 1–3 (for example SD HS Circularity ranging from 0.04–0.049), while the least to classes 8–9 (SD HS Circularity from 0.21–0.23). (3) The most significant changes in the values occurred in the lowest classes (1–2) and gradually decreased to least in the classes representing the greatest roundness of grains (8–9). Thus, it was concluded that morphometric differentiation decreases with an increase in their roundness.

It was found that the boundary values between the parameters calculated for the grains analysed occurred between certain classes distinguished by Krumbein: (1) Between 1 and 2, 8 and 9, the limit was the value of the HS Circularity parameter: 0.91 (Figure 3a); (2) between 1 and 2, 8 and 9, the limit was the value of the Convexity parameter = 0.997 (Figure 3b); (3) between 1 and 2, 6 and 9, the limit was the value of the Solidity parameter = 0.982 (Figure 3c). On the other hand, Aspect Ratio values range from 0.56 to 1 and could not be used to separate any of the Krumbein classes (Figure 3d).

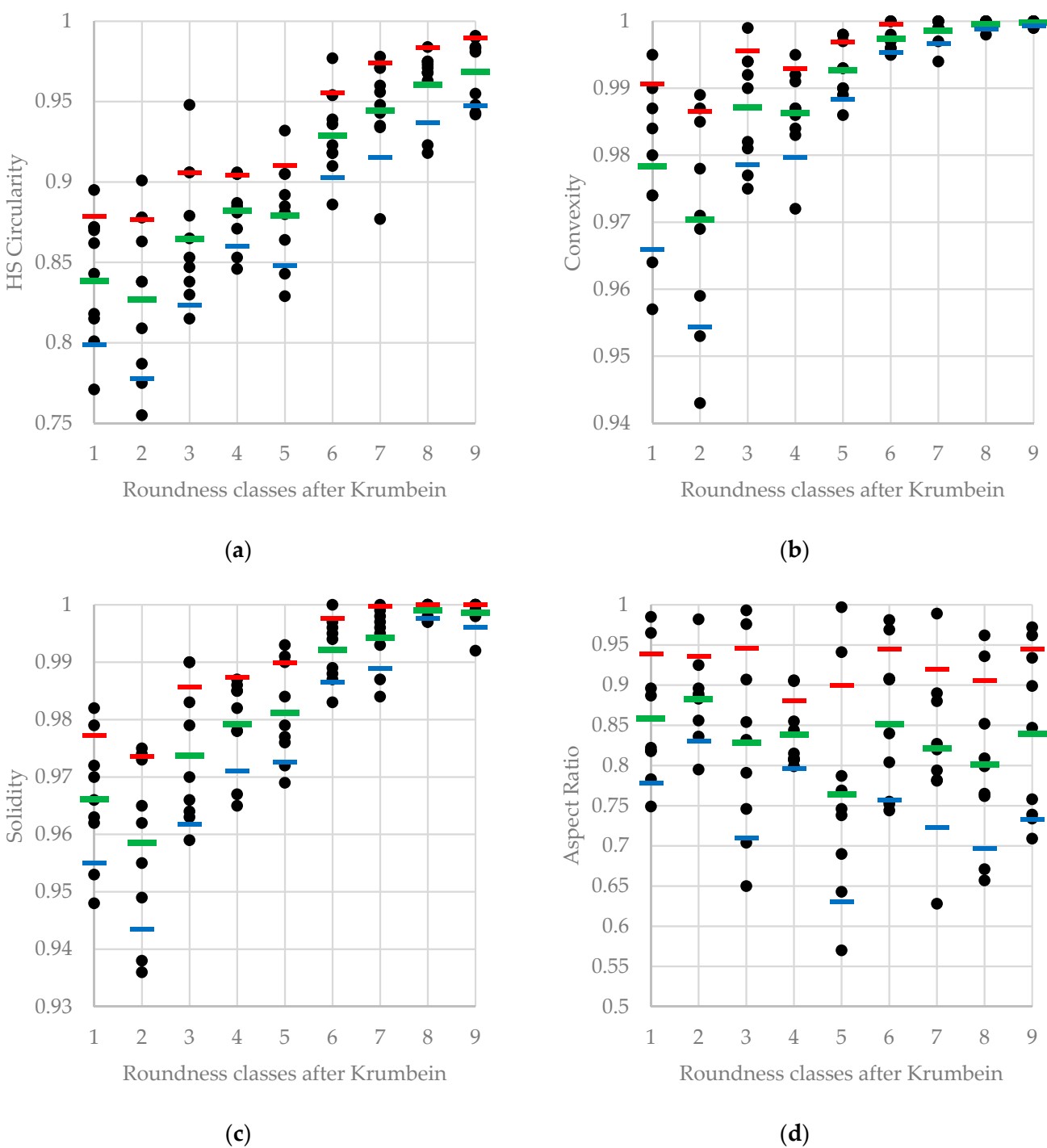

**Figure 3.** Values of patterns of grain roundness classes by Krumbein [4]: (**a**) HS Circularity; (**b**) Convexity; (**c**) Solidity; (**d**) Aspect Ratio. Red marker: AVG + SD, blue marker: AVG − SD, green marker: AVG.

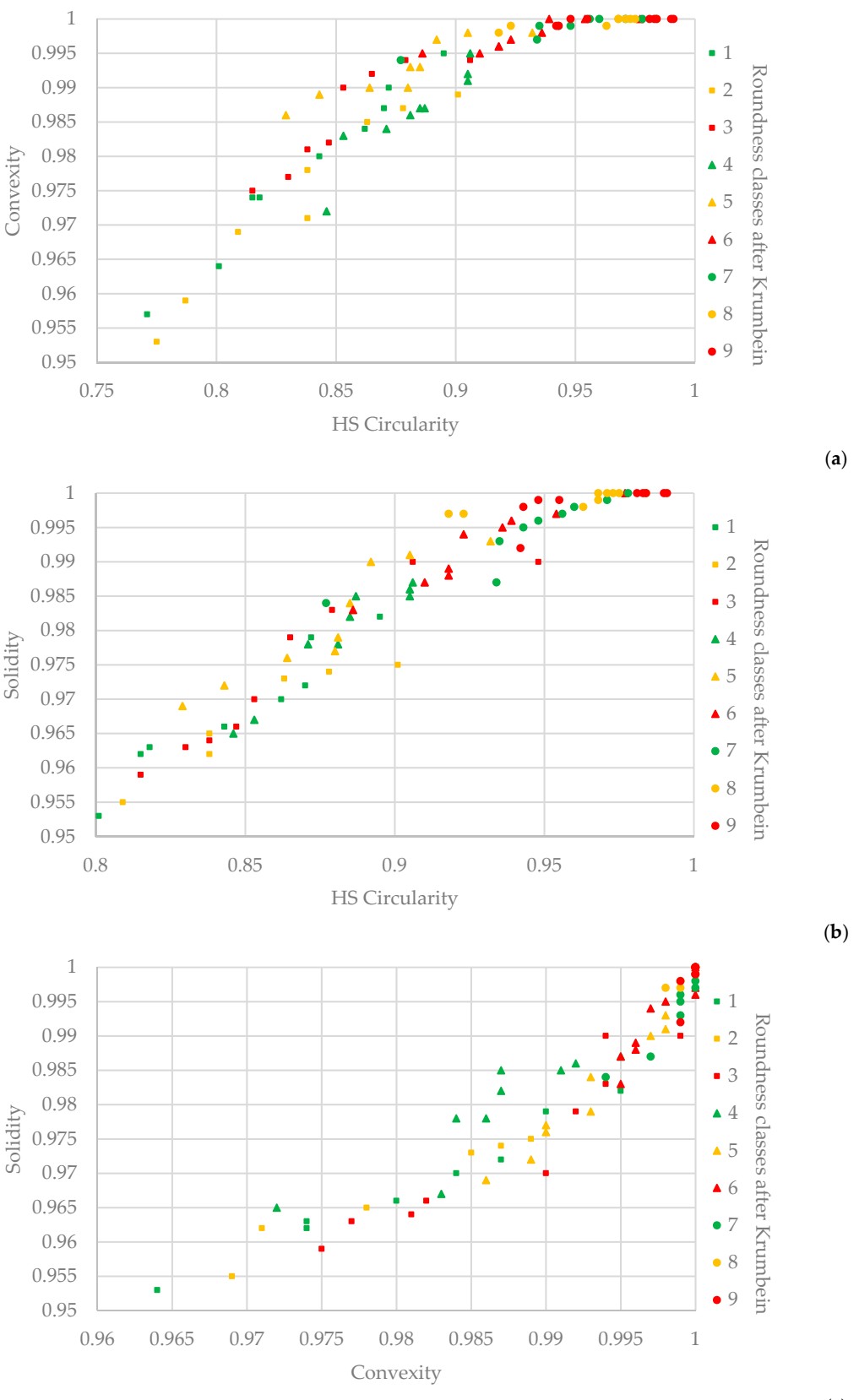

**Figure 4.** Diagrams of dependence of statistical parameters: (**a**) HS Circularity vs. Convexity; (**b**) HS Circularity vs. Solidity; (**c**) Convexity vs. Solidity, 1–9 Krumbein classes.

## 4. Discussion

When considering the results of morphometric measurements, it can be concluded that they do not make it possible to distinguish any of the classes proposed by Krumbein [4]. Therefore, visual measurements cannot be replaced by automatic image analysis performed with the Morphologi G3SE.

On the other hand, the high similarity of the values of the morphometric parameters examined for standard grains from different classes leads to a critical approach to classifying the roundness classes proposed by Krumbein [4]. By analysing the data from the measurements, it was found that the highest values of HS Circularity and Convexity parameters of standard grains belonging to classes 1 and 2 are smaller than the lowest values of those belonging to classes 8 and 9. Moreover, the highest values of the Solidity parameter belonging to classes 1–4 are smaller than the lowest included in classes 8 and 9. The ranges of the parameters overlap. While it is possible to separate them from each other by employing the results of measurements of standard grains belonging to extreme classes, it is impossible to distinguish grains belonging to intermediate classes. It is therefore unsurprising that the visual separation of individual classes using Krumbein's shapes is subjective. Results from evaluating grain roundness in this way may be considered questionable.

However, while analysing the results, an attempt can be made to automatically separate rounded grains from those not rounded (angular). Therefore, the results of morphometric measurements were analysed to define the limit values for individual parameters. The elimination of standard grains with anomalous HS Circularity values caused the limit of the lowest value for the grain shape included in class 7 to be 0.915 (Figure 3a). Therefore, the lowest value of the range of grains typical for class 7 may be one of the criteria for classifying the grains into a rounded class. Moreover, the upper value of the range of grains classified as typical (after eliminating anomalous values) for class 3 was 0.906. It is worth adding that the extreme values of HS Circularity of standard grains in classes 1 and 2 did not exceed 0.901 (Table 1). Thus, the value of HS Circularity (0.906) can be considered the highest for classifying grains as rounded. Both values of the HS Circularity parameter, determined as separating rounded from angular grains, were stricter than those set out in the literature. For example, according to Suzuki et al. [38], the lower limit of grains that can be considered rounded is 0.87. They used the Cox method [39] in their calculations. Similarly, Blott and Pye [37], based on the Riley measurement method [40], concluded from their five-grade circularity classification that the particles with very high circularity/sphericity are in the range 0.894–1.

The value of 0.997 for the Convexity parameter can also be taken as the limit value for rounded grains. Two arguments can support this view: (1) This value separates the highest values in classes 1 and 2 from the smallest in classes 8 and 9; (2) This value marks the limit of negative values for anomalous standard grains from class 7, i.e., AVG − SD: $0.999 − 0.002 = 0.997$ (Table 1). The values of the Solidity parameter characterising the smoothness of the grain surface determine a critical feature of their roundness. As previously stated, the value of this parameter at 0.982 is the border between the highest values of standard grains included in classes 1 and 2 and the lowest values of standard grains included in classes 6–9. This value can be regarded as the highest limit for angular grains. However, considering the shapes and values of the Solidity parameter in Table 1, by analogy to the anomalous limit values of the Convexity parameter of standard grains included in class 7, 0.992 can be assumed to be the lowest limit value for the Solidity of rounded grains.

The Aspect Ratio parameter and the corresponding Elongation parameter, the inverse of the Aspect Ratio [27], should not be used in automated classification concerning Krumbein's formula [4]. These parameters directly depend on grain width to length ratio and do not consider edge unevenness [27]. In all classes of Krumbein's standard chart, there are more elongated and more spherical grains, hence a substantial dispersion of the values found in all classes (Table 1 and Figure 3d). Therefore, it can be concluded that these parameters are useless in an analysis of roundness. A similar conclusion regarding the use

of Aspect Ratio in genetic analyses was drawn by Tuwal et al. [14], and this indicator had the weakest correlation compared with other parameters. Sochan et al. [34] also rejected the elongation parameters in the analysis of grains that were equivalent to Krumbein's classes. The Circularity and Convexity parameters are more critical than the Aspect Ratio in the context of sedimentological studies, as indicated by Campaña et al. [28]. However, the Aspect Ratio and Elongation can be successfully used in analyses by the Zingg method [41], based on grain axis measurements. Such studies were conducted, among others, by Barret [42], Blott and Pye [37], Tarriño [43]. The Aspect Ratio (or Elongation) parameter as a diagnostic feature of sediments is used in grain-transportation distance studies because the grain shape determined by these parameters is modified later than grain roundness [37,44–46].

As already mentioned, it is impossible to define specific morphometric parameters for the particles that would directly refer to the standard grains used for the Krumbein visual method. Problems in automating manual methods on the Morphologi G3SE apparatus have already been pointed out by Becker et al. [47]. They found that the machine's automatic particle counting, divided into organic and inorganic, was not precise, but the results correlated with the results of the manual method. In addition, Resentini et al. [31] stated that, despite technological progress, an entirely satisfactory automatic method for determining the angularity of detrital particles had not yet been found. Despite this, the Morphologi G3SE apparatus is often used in sedimentological studies of grains physically placed in the apparatus [28,32,48–50] and images of mineral particles [30,51,52].

The morphometric indicators of grain shape, especially quartz grains, are the basis for assessing their textural maturity. The results of grain-maturity studies using indicators of roundness, angularity, fractal dimension, irregularity, rectangularity, convexity and solidity by Tuwal et al. [14] showed that angularity and fractal dimensions are the best for assessing sediment maturity. Moreover, the most rounded aeolian grains showed the greatest maturity. Based on the research results presented here, it can be concluded that good results can be obtained in evaluating the maturity of quartz grains using HS Circularity, Convexity and Solidity indicators. These parameters, unlike those used by Tuwal et al. [14], correlate well with each other and, considered together, allow for low textural maturity (young) grains from high textural maturity (old) to be distinguished. Sochan et al. [34], using the Morphologi G3 apparatus, did not analyse Krumbein's charts, but only the corresponding grain shapes. In their conclusions, they stated that none of the shape parameters allowed for the direct categorization of a particle into a particular Krumbein roundness class. They also did not propose any shape parameter values that would be the basis for distinguishing angular from rounded grains. Whereas, as a result of our research, we have indicated the values of three parameters that constitute the criterion for automatic differentiation of grains of these two types.

In the genetic analysis of sediments, especially as a diagnostic feature of aeolian sediments, an important indicator is the matte of the surface of the grains [7,19]. On the Morphologi G3SE apparatus, it is possible to measure the matte of the surface with two indicators: Intensity Mean and Intensity SD [27]. Research on the use of these two indicators in the interpretation of sediment genesis, already carried out by the authors, constitutes the next stage in applying automatic grain-image analysis to sedimentological analyses.

## 5. Conclusions

Based on the morphometric analyses of Krumbein standard grains [4] performed with the Morphologi G3SE apparatus, it was found that they cannot directly replace visual assessment. The main reason is that the automatic measurement results showed a very high similarity between the parameters of standard grains belonging to different classes, not only neighbouring but also distant ones. Small differences between grains mean that particles may be misclassified during subjective visual evaluation.

As a result, we propose limit values for the parameters, which can be considered criteria for the automatic determination of grains as rounded or angular. Grains can be considered rounded whose values for all three parameters, seen as vital in the automatic

analysis with the Morphologi G3SE apparatus, are more significant than HS Circularity—0.915, Convexity—0.997, and Solidity—0.992. However, grains can be considered angular whose values are lower than HS Circularity—0.906, Convexity—0.994, and Solidity—0.983. Moreover, when classifying a grain as spherical, the Aspect Ratio value must be close to 1, considering the 10% difference between grain length and width dimensions. According to Formula (4) [27], this should not be less than 0.9. If the values of the parameters do not meet these conditions, the grain can be described as subrounded.

Although automatic image analysis does not allow for the precise identification of grains belonging to any of the classes separated by the Krumbein method, the values for HS Circularity, Convexity and Solidity parameters can help classify them into three types: angular, subrounded and rounded. On the other hand, the Aspect Ratio or Elongation parameters can be additional criteria for determining grain sphericity.

**Author Contributions:** Conceptualization, J.B.S. and K.W.; methodology, J.B.S. and K.W.; software, K.W.; validation, J.B.S. and K.W.; formal analysis, K.W.; investigation, J.B.S. and K.W.; resources, J.B.S. and K.W.; data curation, J.B.S. and K.W.; writing—original draft preparation, J.B.S. and K.W.; writing—review and editing, J.B.S. and K.W.; visualization, J.B.S.; supervision, J.B.S.; project administration, J.B.S.; funding acquisition, J.B.S. and K.W. All authors have read and agreed to the published version of the manuscript.

**Funding:** This research was funded by Pedagogical University of Krakow Project No. BN.610-334/PBU/2020.

**Acknowledgments:** The authors are grateful to the two anonymous reviewers for their helpful suggestions and comments.

**Conflicts of Interest:** The authors declare no conflict of interest.

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
