# Peer review of "Morphometric Parameters of Krumbein Grain Shape Charts—A Critical Approach in Light of the Automatic Grain Shape Image Analysis"

_minerals, doi:10.3390/min11090937_

Round 1
Reviewer 1 Report
I accepted the invitation to review this paper because I expected a step forward in sedimentary petrography and mineralogy and geo-morphometrical studies.
The authors put an “old problem (which has already been solved to a large extent) in a new jacket”. The morphology of grains covering the full particle-size range from fine-grained sand (even silt) to gravel has always been selected to get an idea of the mode of transport and the depositional environment. Using grain size only, has been proven to do not contribute to a solution of any of these questions even if laser-based camera devices such as CAMSIZER or image-analysis are used. I myself used both methods since several years for both issues mentioned above and tested especially the parameters, area, circularity and roundness which is still the most relevant because it bridges the gap between the two analytical techniques.
The authors state:” An automated quantitative analysis would eliminate subjective judgment unavoidable with non-automated visual analysis”. This cannot be the scope of a scientific approach taken in a journal like MINERALS. Because this method per se can neither cater in the way dealt with for any geoscientific solution nor for material sciences. The positive correlation with different correlation coefficients is not anything out of the ordinary. Nobody will use today for grain size analyses Krumbein´s 1941 paper. Numerical data are common sense. So what is this method useful for ?
The authors must show how they can cater for the needs and wants of sedimentologists (see below) because all of these previous grain size analyses were launched by sedimentologists. Only when they can demonstrate that their approach is a way forward the ms. is worth to be published.
There are numerous numerical approaches taken to provide solutions for sedimentology (see below). It is important to note that any grain size analysis must take into consideration compositional investigations otherwise is will not provide any solution.
DILL, H.G., BUZATU A., BALABAN S.-I. , UFER K., TECHMER A. SCHEDLINSKY, W., and FÜSSL M., (2020) The transition of very coarse-grained meandering to straight fluvial drainage systems in a tectonized foreland-basement landscape during the Holocene (SE Germany) – A joint geomorphological-geological study.- Geomorphology 370: 107364
DILL, H.G., BUZATU A., BALABAN S.-I. , UFER K., GÓMEZ TAPIAS J., BÎRGĂOANU D. and CRAMER T. (2020) The “badland trilogy” of the Desierto de la Tatacoa, Upper Magdalena Valley, Colombia, a result of geodynamics and climate: With a review of badland landscapes.- Catena 194: 1-20.
DILL, H.G., BUZATU A., and BALABAN S.-I. (2021) Coastal morphology and heavy mineral accumulation in an upper-macrotidal environment– A geological-mineralogical approach from source to trap site in a natural placer laboratory (Channel Islands, Great Britain).- Ore Geology Reviews 138: 104311
Please do not use decimal comma but only decimal points.
I suggest major revisions of the paper to comply with the requirements raised above.
Author Response
Thank you very much for all the comments you kindly presented in the reviews. We have made corrections into the original manuscript according to them. In introduction and in discussion, a number of publications on automatic shape analysis were indicated that were not mentioned earlier (Sochan et al. 2015, Tunwal et al. 2018, 2020). Attention was paid to the significance of the grain shape in the genetic interpretation, referring to works e.g.: Mycielska-Dowgiałło and Woronko (1998), Kasser et al. (2007), Woronko et al. (2017); Sandeep et al. (2018),Tunwal et al. (2018). In the justification of the research objective, it was emphasized that so far no one has indicated the values of morphometric indicators calculated by the Morphologi G3SE apparatus, that would constitute the criterion for the separation of angular and rounded grains. This is a new solution which, in our opinion, is an important step in automating the research on the shape of quartz grains. The analysis of the shape of quartz grains is important in the assessment of the textural maturity of the sediment and the sedimentological interpretation of the deposition environments (Tunwal et al 2018, 2020).
It is true that "any grain size analysis must take into consideration compositional investigation". The authors of the article are preparing a publication in which the results of the shape of sand grains transported by wind will be presented. Indeed, the mineral composition of the sediment is of great importance here, which is evident in the comparison of aeolian sediments from the Sahara, Iceland, New Zealand, South America and Europe. However, in the case of quartz grains, the analysis of their shape in relation to Krumbein's chats is still of great interpretative importance. It is worth adding that the importance of grain size composition analyses in sedimentological analyses is questionable, for example, trends in grain size distribution changes in river alluvia can be the basis for many geoscientific interpretations (Szmańda 2018, Dolbunova et al. 2020, Kittel et al. 2020, Szmańda et al., 2021). The particle size analysis data are useful in the estimation of shear and settlement velocities.
Szmańda J.B., 2018, Main determinants of the grain size distribution of overbank deposits in Poland – an overview of literature on models of sedimentation. Geological Quarterly, 62 (4): 873–880. doi: http://dx.doi.org/10.7306/gq.1444
Dolbunova, E., Szmańda, J., Kittel, P., Kulkova M., Aleksandrovskiy A., Cywa, K., Mazurkevich, A., 2020, Rrakushechny Yar site: lacustrine and fluvial deposits, buried soils and shell platforms from 6th mill BC. Acta Geographica Lodziensia, 109: 61-80, https://doi.org/10.26485/AGL/2020/110/5.
Kittel, P., Mazurkevich, A., Alexandrovskiy, A., Dolbunova, E., Krupski, M., Szmańda, J., Stachowicz‐Rybka, R., Cywa, K., Mroczkowska, A., Okupny, D., 2020, Lacustrine, fluvial and slope deposits in the wetland shore area in Serteya, Western Russia. Acta Geographica Lodziensia, 109: 103-124, https://doi.org/10.26485/AGL/2020/110/7.
Szmańda J.B., Gierszewski P.J., Habel M., Luc M., Witkowski K., Bortnyk S., Obodovskyi O., (2021), Response of the Dnieper river fluvial system to the river erosion caused by the operation of the Kaniv hydro-electric power plant (Ukraine), Catena, 202: 105265. https://doi.org/10.1016/j.catena.2021.105265.
We disagree with the opinion that "The positive correlation with different correlation coefficients is not anything out of the ordinary". The results of the research on the correlation of various morphometric indices of grains published by Tunwal et al. (2018) indicate that the relationships between them are positive and negative, and with different strength of the relationship. Therefore, we consider the finding of very strong positive relationships between the three indicators studied by us to be an important basis for using their values as a criterion for automatic analysis of the particle shape.
Changes to the text were introduced in red font.
The manuscript text has also been linguistically revised by native spiker.

Reviewer 2 Report
General Comments:
The paper analyses Krumbein’s visual chart using Morphologie G3SE’s image analysis software. The work presents a way to calibrate quantitative shape analysis parameters with traditionally used visual chart. Selection of the 4 parameters for this study isn’t explained. Since Krumbein visual chart is based on roundness/angularity of particles, the usefulness of aspect ratio for this study needs to be justified. It is expected that particles with low or high aspect ratio can be either angular or rounded. A number of important references are missing from the article which is discussed the in detailed comments.
Detailed comments:
- Line 28: Replace “grain size” with “texture”
- Line 35-36: Mention image analysis software which have been used for quantitative shape analysis in geology. Some of these software includes:
ImageJ: Schneider, C.A., Rasband, W.S., Eliceiri, K.W. "NIH Image to ImageJ: 25 years of image analysis". Nature Methods 9, 671-675, 2012
ImageSXM: Heilbronner, R., Barrett, S., 2014. Image Analysis in Earth Sciences: Microstructures and Textures of Earth Materials. Springer, Heidelberg, p. 513
IPSAT: Tunwal, M., Mulchrone, K.F., Meere, P.A., Image based particle shape analysis toolbox (IPSAT), Computers and Geosciences. 135 (2020), 104391
FSA: Chávez, G.M., Castillo-Rivera, F., Montenegro-Ríos, J.A., Borselli, L., Rodríguez-Sedano, L.A. and Sarocchi, D., 2020. Fourier Shape Analysis, FSA: Freeware for quantitative study of particle morphology. Journal of Volcanology and Geothermal Research, 404, p.107008.
- Line 43-44: Include study by Sochan et al 2015 which used Morphologie G3 instrument for selection of shape parameters. Sochan et al 2015 also used Krumbein’s chart in their work. Explain how the current work is novel in this context. Include Sochan et al 2015’s work in the discussion.
Sochan, A., Zielienski, P. and Bieganowski, A. (2015) Selection of shape parameters that differentiate sand grains, based on the automatic analysis of two dimensional images. Sedimentary Geology, 327, 14–20.
- Line 46-48: Some studies have used multivariate analysis for discriminating different population of sediment grains, for example, Campana et al., 2016 and Tunwal et al., 2018. In yet another approach, multi-parameter approach has been developed for particle shape quantification (Tunwal et al., 2020). Include these points in your introduction/discussion.
Campana et al., 2016 is reference 17 in your text.
Tunwal, M., Mulchrone, K.F., Meere, P.A., 2018. Quantitative characterization of grain shape: implications for textural maturity analysis and discrimination between depositional environments. Sedimentology 65, 1761–1776.
Tunwal, M., Mulchrone, K. F., & Meere, P. A. (2020). A new approach to particle shape quantification using the curvature plot. Powder Technology, 374, 377-388.
- Line 55-58: Rephrase these statements to clearly state your aim.
- Line 82-84: Explain how the selection of these four parameters was performed. Especially, inclusion of aspect ratio needs to be justified. Refer to general comments.
- Line 101: In Figure 2 caption explain what the gray area and dotted line represents in the figure.
- Line 126-132: Discuss how these correlations compare to correlation of shape parameters in the study Tunwal et al., 2018.
- Line 150-152, 223-229: refer to earlier comments regarding aspect ratio.
- Discussions: Include the missing references mentioned above in the discussion section as appropriate.
Author Response
Thank you very much for all the comments you kindly presented in the reviews. We have made corrections into the original manuscript according to them. We especially thank you for the recommendations of the publication by Sochan et al. (2015) and Tunwal et al. (2018), which strengthened the discussion of the research results. The manuscript text has also been linguistically revised by native spiker. Changes to the text were introduced in red font.
- Line 28: Replace “grain size” with “texture”
Response: changed
- Line 35-36: Mention image analysis software which have been used for quantitative shape analysis in geology. Some of these software includes:
ImageJ: Schneider, C.A., Rasband, W.S., Eliceiri, K.W. "NIH Image to ImageJ: 25 years of image analysis". Nature Methods 9, 671-675, 2012
ImageSXM: Heilbronner, R., Barrett, S., 2014. Image Analysis in Earth Sciences: Microstructures and Textures of Earth Materials. Springer, Heidelberg, p. 513
IPSAT: Tunwal, M., Mulchrone, K.F., Meere, P.A., Image based particle shape analysis toolbox (IPSAT), Computers and Geosciences. 135 (2020), 104391
FSA: Chávez, G.M., Castillo-Rivera, F., Montenegro-Ríos, J.A., Borselli, L., Rodríguez-Sedano, L.A. and Sarocchi, D., 2020. Fourier Shape Analysis, FSA: Freeware for quantitative study of particle morphology. Journal of Volcanology and Geothermal Research, 404, p.107008.
- Response: Publications have been included in the text.
- Line 43-44: Include study by Sochan et al 2015 which used Morphologie G3 instrument for selection of shape parameters. Sochan et al 2015 also used Krumbein’s chart in their work. Explain how the current work is novel in this context. Include Sochan et al 2015’s work in the discussion.
Sochan, A., Zielienski, P. and Bieganowski, A. (2015) Selection of shape parameters that differentiate sand grains, based on the automatic analysis of two dimensional images. Sedimentary Geology, 327, 14–20.
and
- Line 46-48: Some studies have used multivariate analysis for discriminating different population of sediment grains, for example, Campana et al., 2016 and Tunwal et al., 2018. In yet another approach, multi-parameter approach has been developed for particle shape quantification (Tunwal et al., 2020). Include these points in your introduction/discussion.
Campana et al., 2016 is reference 17 in your text.
Tunwal, M., Mulchrone, K.F., Meere, P.A., 2018. Quantitative characterization of grain shape: implications for textural maturity analysis and discrimination between depositional environments. Sedimentology 65, 1761–1776.
Tunwal, M., Mulchrone, K. F., & Meere, P. A. (2020). A new approach to particle shape quantification using the curvature plot. Powder Technology, 374, 377-388.
Response: changed to: Genetic interpretation of sediments is often based on the results of sand (quartz) grain-coating analyses. Textural maturity of sediments depends on the shape of grains [18]. This shape, however, depends on the characteristics of the source material and the transport distance [7,19,20]. Regardless of the characteristics of the sedimentation medium (air or water), roundness increases with distance of grain transport [21]. Therefore, in interpreting the origin of sediments, the analysis of the shape of sand grains is essential. However, it should be emphasised that grain shapes in various sedimentary environments are similar, especially fluvial, aeolian and beach sediments, as well as glacial and weathered [14,19].
- Line 55-58: Rephrase these statements to clearly state your aim.
Response: changed to: An automated quantitative analysis would eliminate the subjective judgment un-avoidable with non-automated visual analysis. So far, no values for specific parameters calculated by the Morphologi G3SE software, which would allow distinguishing angular, subrounded and rounded grains, have been determined. Because Krumbein's chart [4] is a model that has repeatedly constituted the basis for distinguishing the types of grain roundness, the question arises if it is possible to automate analysis of Krumbein’s patterns using measurement data from the Morphologi G3SE device. To answer this was the main task of the research. Its most important goal was to define criteria for separating angular, subrounded and rounded grains with morphometric parameter values. The article also forms a part of the discussion that has been going on for many years on the significance of Krumbein grain-shape classes when analysing the coating of mineral clusters (mainly quartz sand grains).
Line 82-84: Explain how the selection of these four parameters was performed. Especially, inclusion of aspect ratio needs to be justified. Refer to general comments.
Response: changed to: The choice of these parameters resulted from their importance in comprehensive analyses of shapes previously performed with this device [28–34,36]. Similar parame-ters were also used in other grain-coating studies [14,28,37,38].
Line 101: In Figure 2 caption explain what the gray area and dotted line represents in the figure.
Response: changed to: Figure 2. Scheme of a particle - the gray area (A) and hull - dotted line (A+B) [16].
- Line 126-132: Discuss how these correlations compare to correlation of shape parameters in the study Tunwal et al., 2018.
and
- Line 150-152, 223-229: refer to earlier comments regarding aspect ratio.
- Discussions: Include the missing references mentioned above in the discussion section as appropriate.
Response: added to the text:
Line 234-238
A similar conclusion regarding the use of Aspect Ratio in genetic analyses was drawn by Tuwal et al. [14], and this indicator had the weakest correlation compared with other parameters. Sochan et al. [34] also rejected the elongation parameters in the analysis of grains that were equivalent to Krumbein's classes.
Line 257-280
The morphometric indicators of grain shape, especially quartz grains, are the basis for assessing their textural maturity. The results of grain-maturity studies using indi-cators of roundness, angularity, fractal dimension, irregularity, rectangularity, convex-ity and solidity by Tuwal et al. [14] showed that angularity and fractal dimensions are the best for assessing sediment maturity. Moreover, the aeolian grains which are best-rounded show the greatest maturity. Based on the research results presented here, it can be concluded that good results can be obtained in evaluating the maturity of quartz grains using HS Circularity, Convexity and Solidity indicators. These param-eters, unlike those used by Tuwal et al. [14], correlate well with each other and, con-sidered together, allow low textural maturity (young) grains from high textural ma-turity (old) to be distinguished. Sochan et al. [34], using the Morphologi G3 apparatus, did not analyse Krumbein’s charts, but the corresponding grain shapes. In their con-clusions, they stated that none of the shape parameters allowed the direct categoriza-tion of a particle into a particular Krumbein roundness class. They also did not propose any shape parameter values that would be the basis for distinguishing angular from rounded grains. Whereas, as a result of our research, we have indicated the values of three parameters that constitute the criterion for automatic differentiation of grains of these two types.
In the genetic analysis of sediments, especially as a diagnostic feature of aeolian sediments, an important indicator is the matte of the surface of the grains [7,19]. On the Morphologi G3SE apparatus, it is possible to measure the matte of the surface with two indicators: Intensity Mean and Intensity SD [27]. Research on the use of these two indicators in the interpretation of sediment genesis, already carried out by the authors, constitutes the next stage in applying automatic grain-image analysis to sedimentological analyses.

Round 2
Reviewer 1 Report
Dear authors,
the side effects influencing the methods presented and the constraints have now been referred to in the revised manuscript
Author Response
Dear Reviewer,
Thank you for accepting the changes to the.
Reviewer 2 Report
Some minor edits are required:
Line 66-68: Add references for the said discussion.
Line 129: Rectify spelling to "Length"
Line 235 and thereafter in the text: Rectify spelling of reference to "Tunwal et al."
Author Response
Dear Reviewer,
Thank you for valuable and constructive comments,
minor changes were made.
Line 66-68: Add references for the said discussion.
Response: changed to: The article also forms a part of the discussion that has been going on for many years on the significance of Krumbein grain-shape classes when analysing the coating of miner-al clusters (mainly quartz sand grains) [7,8,14,19,28,31,34-36].
Line 129: Rectify spelling to "Length"
Response: changed
Line 235 and thereafter in the text: Rectify spelling of reference to "Tunwal et al."
Response: changed
